# Genome-Wide Association between the 2q33.1 Locus and Intracranial Aneurysm Susceptibility: An Updated Meta-Analysis Including 18,019 Individuals

**DOI:** 10.3390/jcm8050692

**Published:** 2019-05-16

**Authors:** Eun Pyo Hong, Bong Jun Kim, Jin Pyeong Jeon

**Affiliations:** 1Molecular Neurogenetics Unit, Center for Genomic Medicine, Massachusetts General Hospital, Boston, MA 02114, USA; ephong0305@gmail.com; 2Broad Institute of MIT and Harvard, Cambridge, MA 02142, USA; 3Institute of New Frontier Stroke Research, Hallym University College of Medicine, Chuncheon 24252, Korea; luckykbj@naver.com; 4Department of Neurosurgery, Hallym University College of Medicine, Chuncheon 24252, Korea; 5Genetic and Research Inc., Chuncheon 24252, Korea

**Keywords:** 2q33.1, intracranial aneurysm, multi-ethnic meta-analysis, subarachnoid hemorrhage

## Abstract

Previous genome-wide association studies did not show a consistent association between the *BOLL* gene (rs700651, 2q33.1) and intracranial aneurysm (IA) susceptibility. We aimed to perform an updated meta-analysis for the potential IA-susceptibility locus in large-scale multi-ethnic populations. We conducted a systematic review of studies identified by an electronic search from January 1990 to March 2019. The overall estimates of the “G” allele of rs700651, indicating IA susceptibility, were calculated under the fixed- and random-effect models using the inverse-variance method. Subsequent in silico function and cis-expression quantitative trait loci (cis-eQTL) analyses were performed to evaluate biological functions and genotype-specific expressions in human tissues. We included 4513 IA patients and 13,506 controls from five studies with seven independent populations: three European-ancestry, three Japanese, and one Korean population. The overall result showed a genome-wide significance threshold between rs700651 and IA susceptibility after controlling for study heterogeneity (OR = 1.213, 95% CI: 1.135–1.296). Subsequent cis-eQTL analysis showed significant genome-wide expressions in three human tissues, i.e., testis (*p* = 8.04 × 10^−15^ for *ANKRD44*), tibial nerves (*p* = 3.18 × 10^−10^ for *SF3B1*), and thyroid glands (*p* = 4.61 × 10^−9^ for *SF3B1*). The rs700651 common variant of the 2q33.1 region may be involved in genetic mechanisms that increase the risk of IA and may play crucial roles in regulatory functions.

## 1. Introduction

The term unruptured intracranial aneurysm (UIA) refers to a localized outpouching of the cerebral artery due to a defect in the muscular layer [1,2]. The incidence of UIAs has been reported to be 15.6 per 100,000 persons in the United States [3]. In most cases, patients with UIA are asymptomatic and are diagnosed incidentally during health check-ups. However, if a subarachnoid hemorrhage (SAH) due to an IA rupture occurs, mortality rates up to 44%, early after the rupture, have been reported [4]. Even after successful recoveries from SAHs, the survivors showed 32% higher mortality rates 30 years later, compared to the general population [4]. Accordingly, numerous clinical, epidemiologic, and genetic studies have been conducted to identify the prognostic determinants for IA formation and rupture [5,6,7]. Genome-wide association studies (GWASs) and meta-analyses have been conducted to discover genetic variations related to IA development in multi-ethnic populations. Candidate loci have included *EDNRA* (rs6841581, 4q31.23), *SOX17* (8q11.23), *CDKN2B-AS1* (9p21), *CNNM2* (10q24.3), *FGD6* (12q22), *STARD13* (13q13), and *RRBP1* (20p12.1) [8,9,10]. These candidate single-nucleotide polymorphisms (SNPs) associated with IAs may indicate that abnormalities in the extracellular matrix (ECM) and vascular endothelium may be involved in the biological mechanism of IA [10].

In previous GWASs, the association of rs700651, mapped to several genes with IA, such as *BOLL*, *PLCL1, SF3B1*, and *ANKRD44* within the 2q33.1 region, was underpowered in multi-ethnic populations [9]. Specifically, the Japanese population somehow lacked this association (odds ratio (OR) = 1.065; 95% confidence interval (CI): 0.992–1.142) compared with European-ancestry populations (OR = 1.186, 95% CI: 1.092–1.267) [10]. While a recent GWAS by Hong et al. [11] reported 29 novel candidate IA susceptibility genes, including *GBA, ARHGAP32, OLFML2A*, and *SCARF1*, the authors also reported two variants in Korean adults, rs6841581, near the 5′-untranslated region (UTR) of *EDNRA* (*p* = 6.5 × 10^−4^), and rs700651, in the intron of *BOLL* (*p* = 0.008). Although ethnic differences in genetic susceptibility are known to exist [12], sample size is important in genetic association studies designed to detect a common variant in complex human diseases [13]. Thus, we conducted an updated meta-analysis of studies encompassing ethnic differences to increase the statistical power for rs700651 mapped on 2q33.1.

## 2. Materials and Methods 

### 2.1. Literature Search and Data Extraction

PubMed, Embase, and the HuGE Literature Finder electronic databases were searched for studies from January 1990 to March 2019 using the following keywords: “aneurysm(s)”, “intracranial aneurysm(s)”, “subarachnoid hemorrhage”, “genetic association”, “*BOLL*”, “boule homolog, RNA binding protein”, “*PLCL1*”, “phospholipase C like 1 (inactive)”, “rs700651”, “genetic”, and “GWAS”. The inclusion criteria for the studies were as follows: 1) saccular aneurysm studies, 2) studies for candidate genes or GWAS, and 3) IA studies in human subjects. The exclusion criteria were as follows: 1) animal studies; 2) dissection or fusiform aneurysms; 3) infection or traumatic aneurysms; 4) concomitant aneurysms with other cerebrovascular diseases, such as moyamoya disease or arteriovenous malformations; 5) studies without clear information on genotypes, OR, and 95% CI; 6) reviews, meta-analyses, or case reports. 

### 2.2. Meta-Analysis of rs700651 (2q33.1) in Multi-Ethnic Populations

The information on rs700651 analyzed under an additive genetic model, such as allele frequency, ORs, 95% CIs, *p*-values, and covariates adjusted in the multivariate logistic regression model, were collected from seven association studies. Additional subgroup meta-analyses were performed according to ethnicity. The analyses were conducted using the Genome-Wide Association Meta-Analysis (GWAMA) software, (http://www.well.ox.ac.uk/gwama/) [14]. The overall estimates were calculated under the fixed- or random-effect models based on the inverse variance method. The Cochran Q test was performed to estimate I^2^ statistics in order to detect potential publication biases across the individual studies. Forest plots were drawn to describe the previous results for rs700651 obtained from each study, as well as the overall effect sizes under the additive-effect model. Funnel plots were drawn to visualize potential publication biases. Those plots were constructed using the STATA software v.11.2 (Stata Corp., College Station, TX, USA).

### 2.3. In Silico Functional Analysis of rs700651 (2q33.1) in East Asians 

We further investigated rs700651 and its neighbor variants using in silico functional analysis with fine mapping for the 2q33.1 region. Pairwise linkage disequilibrium (LD, r^2^) was estimated for variants within the rs700651 variant position (198,631,714 bp) ± 500,000 bp to detect transcription factor binding sites (TFBSs), splicing sites, and coding variants strongly correlated with this variant. Furthermore, the regional plot was used to identify 1) variants with putative regulatory potential via the RegulomeDB scoring system [15] and 2) minor allele frequencies (MAF). All systematic estimations for the East-Asian populations (Japanese (JPT) and Han Chinese (CHB)) were based on the 1000 Genome Phase 3 dataset of the LDproxy of the LDlink v3.4.0. (https://ldlink.nci.nih.gov/) [16].

### 2.4. Genotype cis-Expression Quantitative Trait Loci (cis-eQTL) Analysis for rs700651 in Human Tissues

In subsequent analyses, we evaluated the genotype-specific expression (GSE) of rs700651 in 48 human tissues via cis-expression quantitative trait loci (cis-eQTL) analysis through the Genotype-Tissue Expression (GTEx, release v7 and human genome build 37) portal (https://gtexportal.org/home/) [17]. Violin plots of the GSE were constructed to visualize normalized gene expressions between three variant genotypes (GG, GA, and AA; G and A alleles indicate the reference (minor) and alternative (major) allele types, respectively). We accounted for a genome-wide significance threshold *p*-value of 5 × 10^−8^ and a suggestive significance threshold *p*-value of 1 × 10^−5^ in the cis-eQTL analysis, which are commonly used in GWASs.

## 3. Results

### 3.1. Identification of Relevant Studies and Study Characteristics

Among nine eligible studies, four studies were excluded from the final analysis because they lacked the outcomes of interest [18] or were meta-analyses [10] or review studies [8,19]. Finally, a total of seven independent cohort studies in five articles were selected for the meta-analysis: Two European GWASs including Finnish and Dutch populations [20]; two Japanese GWASs [5,20]; one European-ancestry candidate gene study including populations from North America, New Zealand, and Australia [21]; one Japanese candidate gene study [22]; and one Korean GWAS (Figure 1) [11]. Among the studies, Bilguvar et al. [20] included three independent cohorts (874/706/495 IA patients and 944/5332/676 controls in Finnish, Dutch, and Japanese populations, respectively). Consequently, this meta-analysis included 4513 patients with IA and 13,506 controls (Table 1).

### 3.2. Meta-Analysis of rs700651 (2q33.1) in Multi-Ethnic Populations

Estimates of the multi-ethnic meta-analysis for rs700651 (2q33.1) were described under the fixed- and random-effect inverse variance models shown in Table 2. Under the random-effect model, studies originating from three European-ancestry and four East-Asian populations, showed an association between rs700651 and an increased risk for IA (OR = 1.164, 95% CI: 1.055–1.284; *p* = 0.0025, Figure 2A), with heterogeneity across the studies (Figure 2B), in particular, in one Japanese study [5]. Without one heterogeneous study, the association had more significance (OR = 1.213, 95% CI: 1.135–1.296; *p* = 1.05 × 10^−8^), with minimal heterogeneity (Table 2). In the ethnicity-based subgroup meta-analysis, patients with IA from the three European-ancestry populations [20,21] showed more significant association with rs700651 under the fixed-effect model (OR = 1.186, 95% CI: 1.092–1.287; *p* = 4.83 × 10^−5^) compared with a meta-analysis of East-Asians with controlled population heterogeneity (I^2^ = 0%, *p* = 0.3749) (Table 2), which contained two Japanese studies [20,22] and one Korean [11] study (OR = 1.264, 95% CI: 1.131–1.413; *p* = 3.60 × 10^−5^). However, the meta-analysis of the overall East-Asian populations did not demonstrate a statistically significant association with IA (OR = 1.179, 95% CI: 0.999–1.392, *p* = 0.0513) (Figure 2C), with a substantial heterogeneity of 76.23% (Figure 2D).

### 3.3. In Silico Functional Analysis of rs700651 in East Asians

Out of a total of 1599 variants within the 1Mbp region, 228 variants showed a pairwise LD with rs700651 (0.8 < r^2^) and MAFs greater than 40% in the East-Asian populations (Figure 3 and Appendix A). These linked variants formed a large haplotype structure on the 2q33.1 region spanning multiple genes, including *BOLL*, *PLCL1*, *RFTN2*, *SF3B1*, and *ANKRD44*. Out of these variants, 141 variants were predicted to be potential regulatory sites (RegulomeDB scores 1f, 2b, 3a, 4, 5, and 6 were seen in 16, 3, 7, 9, 38, and 68 variants, respectively) (see detail in Appendix A). This implies that this large haplotype structure containing regulatory sites might be strongly correlated with genetic modifications conferring susceptibility to IA formation and pathogenesis in East-Asian populations. Furthermore, the common variant of rs1064213 (MAF = 27.54%) in the *PLCL1* gene had an amino acid substitution (Val667Ile, GTA > ATA) which showed LD with rs700651 (r^2^ = 0.2894).

### 3.4. Genotype cis-Expression Quantitative Trait Loci (cis-eQTL) Analysis of rs700651 in Human Tissues

Out of the total 618 genotypic cis-eQTL results for rs700651, three cis-eQTLs reached a genome-wide significance threshold in two eQTL genes, *ANKRD44* (*p* = 8.04 × 10^−15^ in testis tissue) and *SF3B1* (*p* = 3.18 × 10^−10^ in tibial nerve, and 4.61 × 10^−9^ in thyroid tissue; Figure 4 and Appendix A). Seven cis-eQTLs showed suggestive significance in seven tissues (*p* < 1 × 10^−5^). In particular, 8 of 10 genome-wide and/or suggestive cis-eQTLs were downregulated in multiple tissues (−0.59 < slope < −0.12), with the exception of two suggestive cis-eQTL *PLCL1* gene expressions (slope = 0.141 and 0.163 in skeletal muscle and transformed fibroblast cells, respectively). However, the *BOLL* gene, that directly contains this variant in the intron region, showed an insignificant cis-eQTL in human tissues (*p* = 0.1589 in skeletal muscle and *p* = 0.6318 in testis tissues; Appendix A). Additional results of the tissue-specific cis-eQTL for other variants near or located on five main candidate genes mapped on 2q33.1, namely, *BOLL*, *PLCL1*, *RFTN2*, *SF3B1*, and *ANKRD44*, are described in the Appendix A (*p* < 0.01). 

## 4. Discussion

To the best of our knowledge, this study is the first updated meta-analysis on the association between the 2q33.1 region (rs700651) and IA in a large-scale multi-ethnic population containing 18,019 individuals. Overall, the “G” allele of rs700651 consistently increased the risk of IA development in both European-ancestry and East-Asian populations. We confirmed the effect of ethnic similarity on this variant, including shared MAFs and overall effect sizes, in this study. Specifically, the overall results reached a genome-wide significance threshold in our meta-analysis controlled for study heterogeneity (*p* = 1.05 × 10^−8^). 

The *BOLL* gene, belonging to the Deleted in Azoospermia *(DAZ)* gene family, is involved in germ cell development. Accordingly, it functions in the reproductive system, in particular, in human spermatogenesis and infertility [23,24]. Aberrant DNA methylation of the *BOLL* promoter has been observed in patients with colorectal or lung cancers. Kang et al. [23] reported that methylation of the *BOLL* promoter was enhanced in colorectal cancer tissue compared to normal control tissues. In addition, xenograft mice produced by engraftment with the human *BOLL* gene showed significantly higher tumor volumes after subcutaneous implantation, indicating an oncogenic role for *BOLL*. However, the role of *BOLL* in the development of IA is not well understood. A previous study found that females experienced higher aneurysm growth and subsequent rupture after menopause than premenopausal women, suggesting a possible hormonal effect on IA pathogenesis [25]. Kubo et al. [26] reported that the female gender was an independent risk factor for IA development in elderly patients 70 years of age or older. A study demonstrated that the greater proportion of IA seen in female compared to male patients over 50 years old was due to a decrease in estrogen levels [25]. Furthermore, smoking and alcohol use, as well as hypertension, are thought to be associated with an increase in IA in older men [25]. Yeap et al. [27] reported that lower levels of free testosterone and higher levels of luteinizing hormone were independent risk factors for abdominal aortic aneurysms. They suggested that impaired gonadal function may be involved in the pathogenesis of arterial dilatation in older men [27]. Accordingly, the hormonal effects and gonadal dysfunction associated with *BOLL* encoded by the 2q33.1 region should be evaluated further. 

The initial GWAS of IA conducted in 2008 suggested a significant association between the “G” allele of rs700651 and IA in European and Japanese populations [20]. However, the association of the “G” allele in rs700651 with IA susceptibility has demonstrated different statistical significance according to ethnicity. There seemed to be a genetic difference within populations, particularly within Japanese populations, in this study. This heterogeneity was derived from the largest Japanese cohort which included 6823 individuals [5]. Nevertheless, our findings are supported by RegulomeDB scores in silico functional analysis that showed a potential regulatory locus harboring variants in 2q33.1. Particularly, finding the cis-eQTL genotypes implicated the rs700651 variant as a transcriptional regulatory factor.

There are some limitations in this study. The MAF of the “G” allele of rs700651 showed an approximate twofold difference in both European-ancestry and East-Asian populations (MAF = 0.2674 and 0.4614, respectively), confirmed in the 1000 Genome Phase 3 reference panel. Despite the fact that the allele frequency difference in the general populations and a potential ethnic difference in human genome structures were present, we demonstrated a disease-specific genetic similarity in the ethnic populations tested, which showed the same effect direction of this variant susceptibility to IA. Our ethnic subgroup meta-analysis showed a lack of discriminative power to determine IA susceptibility variation in ethnically stratified GWASs due to the small effect size and the limited sample size of some populations, such as those of European-ancestry, East-Asian, and other populations. Nevertheless, our overall results highlighted strong power and a high genome-wide significance threshold (*p* = 1.05 × 10^−8^). The 2q33.1 locus, a region harboring potential regulatory elements, showed regulomeDB scores in in silico functional analysis related to genetic mechanisms of IA formation via fine mapping. In addition, genotypic cis-eQTLs for rs700651 mapped to the regulatory regions of *PLCL1, SF3B1*, and *ANKRD44* showed strong expressions in multiple human tissues, such as testis, tibial nerves, and thyroid glands (*p* < 5 × 10^−8^). If more studies are included in further meta-analyses, the reliability of this locus as a biomarker for IA susceptibility in multi-ethnic populations, as well as in ethnic subgroups, may be confirmed.

## 5. Conclusions

This meta-analysis highlights how an increase in the statistical power of rs700651 mapped to the 2q33.1 region played an important role in identifying regulatory mechanisms, including the positive regulation of translational initiation (*BOLL*) and the regulation of synaptic transmission (*PLCL1*), related to IA susceptibility. Nonetheless, mapping heritability to a point variation is challenging, because single genetic variations cannot account for all heritable IAs and because cerebral aneurysms are affected by multifactorial interactions and components. Thus, the development of multiple polygenic models composed of IA-predicting variants should be pursued in future studies. From this point of view, our findings may offer better genetic information for IA diagnoses and provide a foundation for the next step in understanding IA pathogenesis.

## Figures and Tables

**Figure 1 jcm-08-00692-f001:**
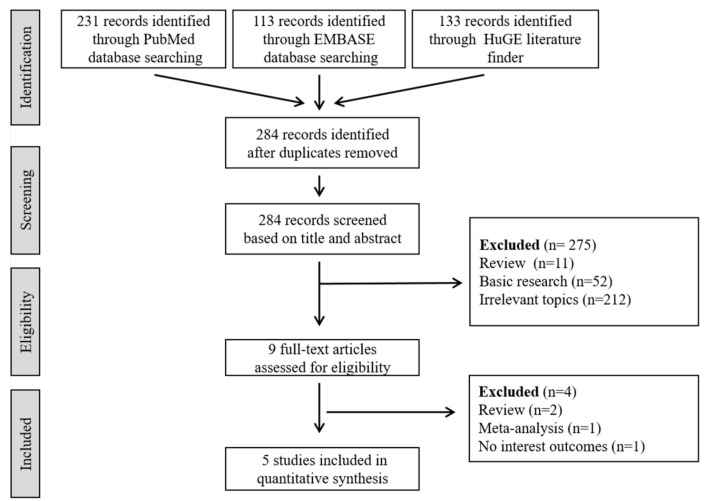
Flow chart of the systematic review of previous genetic/genome-wide association studies.

**Figure 2 jcm-08-00692-f002:**
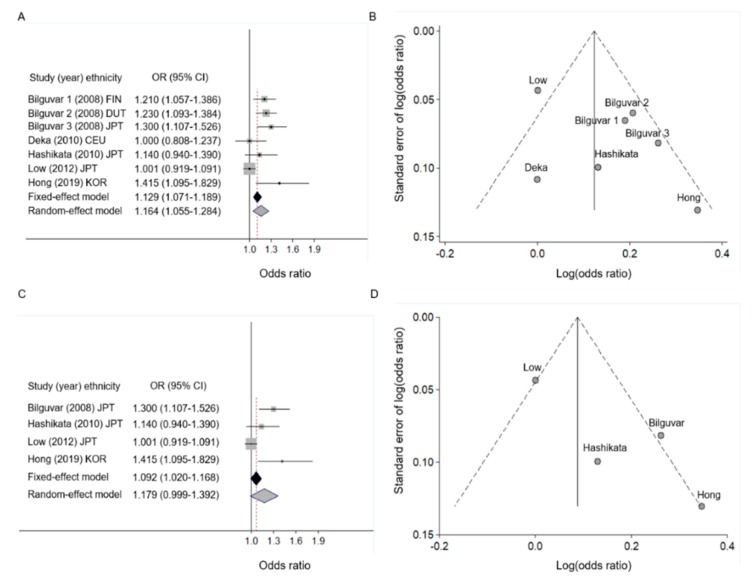
Forest (**A**,**C**) and funnel (**B**,**D**) plots for rs700651 in multi-ethnic meta-analysis and subgroup meta-analysis of East Asians using an inverse-variance model. The first author is indicated in panels (**A**–**D**). The study by Bilguvar et al. 1–3 (2008) [20] contained three genome-wide association studies, including Finnish (FIN), Dutch (DUT), and Japanese (JPT) populations. The study by Deka et al. (2010) [21] included 406 sporadic Caucasian (CEU) patients and 392 healthy controls enrolled from 26 multi-centers. The horizontal line indicates the 95% confidence interval, the shaded square box indicates the weight of each study, and the black spot in the center of the square box indicates the odds ratio of each study. The pooled odds ratio estimates from the fixed-effect (black diamond) or random-effect (grey diamond) inverse-variance models are shown in panels (**A**) and (**C**). The X- and Y-axes in panels (**B**) and (**D**) indicate the pooled log-transformed odds ratio “Log (odds ratio)” and the standard error, respectively.

**Figure 3 jcm-08-00692-f003:**
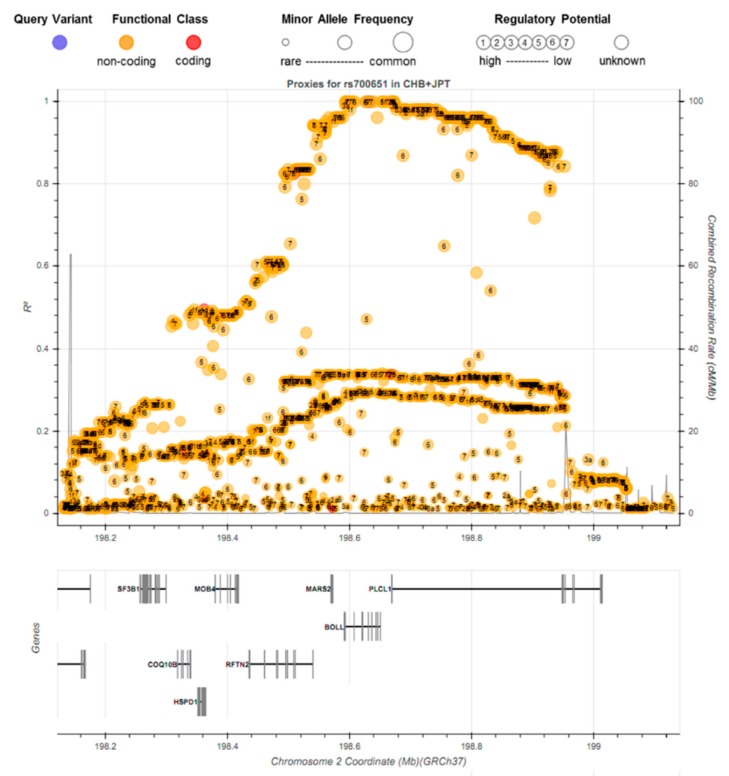
Regional visualization in pairwise linkage disequilibrium (LD, r^2^), functional annotation, RegulomeDB scores, and minor allele frequency for 1599 variants within the rs700651 region variant position ± 500 kbp in East-Asian populations (JPT + CHB of 1000 Genome Phase 3 reference panel).

**Figure 4 jcm-08-00692-f004:**
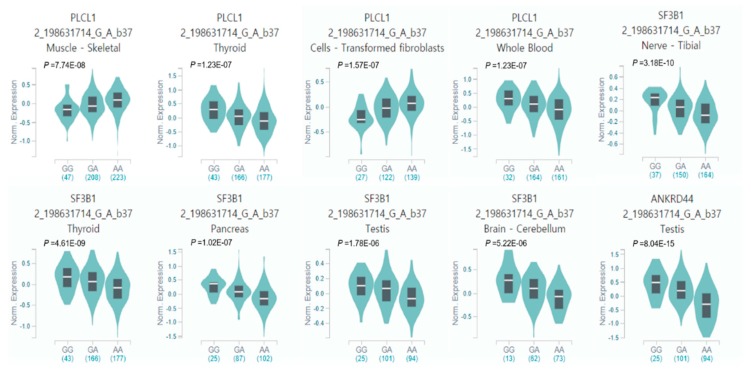
Violin plots of allele-specific cis-eQTLs according to rs700651 genotypes (2_198631714_G_A_b37) in human tissues in the Genotype-Tissue Expression (GTEx, release v7 and human genome build 37) database (*p* < 1 × 10^−6^). G and A alleles indicate the reference (minor) and alternative (major) allele types, respectively. The teal region indicates the density distribution of the samples in each genotype. The white line in the box plot (black) shows the median value of the expression of each genotype.

**Table 1 jcm-08-00692-t001:** Study characteristics of the rs700651 variant in multi-ethnic populations.

Study First Author (Year)	Population	Case/Control	Female	Age ^a^	N/R ^b^	RAF ^b^	OR (95% CI) ^c^	*p*	Adjusted Covariates
Bilguvar (2008) [20]	FIN	874/944	NA	NA	A/G	0.390/0.440	1.210 (1.057–1.386)	0.0058	NA
	DUT	706/5332	NA	NA	A/G	0.350/0.400	1.230 (1.093–1.384)	5.8 × 10^−4^	NA
	JPT	495/676	NA	NA	A/G	0.540/0.460	1.300 (1.107–1.526)	0.0011	NA
Deka (2010) [21]	CEU	406/392	53.3 (45.7%)	50.5/63.4	A/G	0.340/0.339	1.000 (0.808–1.237)	0.973	age
Hashikata (2010) [22]	JPT	419/408	66.1 (52.0%)	60.5/60.0	A/G	0.484/0.463	1.140 (0.940–1.390)	0.19	sex, age, smoking, HTN
Low (2012) [5]	JPT	1359/5464	64.7 (42.7%)	60.1/56.9	A/G	0.490/0.488	1.001 (0.919–1.091)	0.975	sex, age, PCs
Hong (2019) [11]	KOR	254/290	58.4 (52.0%)	59.3/52.1	A/G	0.476/0.449	1.415 (1.095–1.829)	0.0079	sex, age, smoking, HTN

CEU: Caucasian; DUT: Dutch; FIN: Finnish; GWAS: genome-wide association study; HLP: hyperlipidemia; HTN: hypertension; JPT: Japanese in Tokyo; KOR: Korean; NA: not available; PC: principal component; ^a^ Mean age of the case (left) and control (right) groups; ^b^ N/R: non-risk/risk allele type; RAF: risk allele frequency in the case (left) and control (right) groups; ^c^ odds ratio (OR), 95% confidence interval (CI), and *p*-values were estimated by multivariate logistic regression after adjustment for study-specific covariates under an additive effect model.

**Table 2 jcm-08-00692-t002:** rs700651 (2q33.1) in multi-ethnic meta-analysis.

No. Studies ^a^	Population	Effect Model	RAF ^b^	OR (95% CI) ^c^	*p*	Heterogeneity (I^2^), %	*p* for Heterogeneity
*European Ancestry*							
3 [20,21]	3 EUR	Fixed	0.403	1.186 (1.092–1.287)	4.83 × 10^−5^	31.26	0.2335
		Random		1.175 (1.061–1.302)	0.0020		
*East-Asian*							
3 [5,20,22]	3 JPT	Fixed	0.482	1.071 (0.998–1.149)	0.0571	76.16	0.0151
		Radom		1.128 (0.952–1.337)	0.1624		
4 [5,11,20,22]	3 JPT, 1 KOR	Fixed	0.480	1.092 (1.020–1.168)	0.0112	76.23	0.0055
		Random		1.179 (0.999–1.392)	0.0513		
3 [11,20,22]	2 JPT, 1 KOR	Fixed, random	0.459	1.264 (1.131–1.413)	3.60 × 10^−5^	0.00	0.3749
*European + East-Asian*							
7 [5,11,20,21,22]	3 EUR, 3 JPT, 1 KOR	Fixed	0.443	1.129 (1.071–1.189)	5.57 × 10^−6^	66.37	0.0066
		Random		1.164 (1.055–1.284)	0.0025		
6 [11,20,21,22]	3 EUR, 2 JPT, 1 KOR	Fixed	0.415	1.213 (1.135–1.296)	1.05 × 10^−8^	12.35	0.3361
		Random		1.212 (1.128–1.302)	1.65 × 10^−7^		

^a^ The number of studies involved in each meta-analysis*;*
^b^ The average RAF in the control group was calculated using the Genome-Wide Association Meta-Analysis (GWAMA) software*;*
^c^ odds ratio (OR), 95% confidence interval (CI), *p*-values, heterogeneity (I^2^ statistic), and heterogeneity *p*-values were estimated for the East-Asian meta-analyses under the fixed- and random-effect models using the inverse-variance method.

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
