# Peer review of "Genome-Wide Association between the 2q33.1 Locus and Intracranial Aneurysm Susceptibility: An Updated Meta-Analysis Including 18,019 Individuals"

_jcm, 2019, doi:10.3390/jcm8050692_

Round 1

Reviewer 1 Report

The authors of this manuscript performed a meta-analyses in multi-ethnic populations to investigate the effects of rs700651 on 2q33.1 with intracranial aneurysm susceptibility. 

I have several comments to improve this manuscript.

1. In table 1, the RAF column is misleading. For example: in the study of Bilguvar et al., the Case/Control RAF is 0.390/0.440, while in Hong et al., is 0.476/0.449, the association of this SNP in these 2 populations is apparently in the opposite direction. 

the authors should stated how the OR is calculated (which allele is set as reference) before performing meta-analysis. Especially determining the correct risk allele. 

2. The authors should describe the multiple testing/threshold that they used for the eQTL analysis. 

3. In addition to rs700651, the authors might want to consider investigating more variants around 2q33.1 for cis-eQTL analysis. 

4. It is important to put in a statement to emphasize the allele frequency differences of this SNP in different populations in general. 

Author Response

Comment 1: In table 1, the RAF column is misleading. For example: in the study of Bilguvar et al., the Case/Control RAF is 0.390/0.440, while in Hong et al., is 0.476/0.449, the association of this SNP in these 2 populations is apparently in the opposite direction.

The authors should stated how the OR is calculated (which allele is set as reference) before performing meta-analysis. Especially determining the correct risk allele.

Answer: Thank you for your careful review. We reconfirmed the allele frequencies and effect sizes of the risk (R, effect=G) and non-risk (N, reference=A) allele types in Table 1 (see footnote “b” in the paper) (References 1 and 2 below). Previous studies (Reference 1 and 2, below) have focused on the risk of the (effect/minor) “G” allele, suggesting positive odds ratios and 95% confidence intervals in independent populations. If the effect allele was “A”, that would represent major allele frequencies in the cases and control groups. For example, 1-0.390 = 0.610 in the cases and 1-0.440 = 0.560 in the control for the Finnish population (Reference 1 below) and 1-0.476 = 0.524 in the cases and 1-0.449 = 0.551 in the controls for the Korean population (Reference 2 below).

References

1.           Bilguvar, K.; Yasuno, K.; Niemela, M.; Ruigrok, Y.M.; von Und Zu Fraunberg, M.; van Duijn, C.M.; van den Berg, L.H.; Mane, S.; Mason, C.E.; Choi, M.; Gaál, E.; Bayri, Y.; Kolb, L.; Arlier, Z.; Ravuri, S.; Ronkainen, A.; Tajima, A.; Laakso, A.; Hata, A.; Kasuya, H.; Koivisto, T.; Rinne, J.; Ohman, J.; Breteler, M.M.; Wijmenga, C.; State, M.W.; Rinkel, G.J.; Hernesniemi, J.; Jääskeläinen, J.E.; Palotie, A.; Inoue, I.; Lifton, R.P.; Günel, M.  Susceptibility loci for intracranial aneurysm in european and japanese populations. Nat Genet. 2008,40,1472-1477.

2.          Hong, E,P.; Kim, B.J.; Cho, S.S.; Yang, J,S.; Choi, H.J.; Kang, S.H.; Jeon, J.P. Genomic variations in susceptibility to intracranial aneurysm in the korean population. J Clin Med. 2019,8

Comment 2: The authors should describe the multiple testing/threshold that they used for the eQTL analysis.

Answer: Per your recommendation, we provided the threshold in the Materials and Methods section as described below.

We accounted for a genome-wide significance threshold p-value of 5×10-8 and a suggestive significance threshold p-value of 1×10-5 in the cis-eQTL analysis, which are commonly used in GWASs (revised manuscript page 3, lines 108-110).

Comment 3: In addition to rs700651, the authors might want to consider investigating more variants around 2q33.1 for cis-eQTL analysis.  

Answer:  Per your recommendation, additional results of the tissue-specific cis-eQTL for other variants near or located on five main candidate genes mapped on 2q33.1, such as BOLL, PLCL1, RFTN2, SF3B1, and ANKRD44, are described in the Supplementary Table S4,  entitled “Genotype cis-expression quantitative trait loci analysis for variants located on 1 Mbp ± each target gene region around 2q33.1 in 48 human tissues obtained from GTEx dataset (release v7) (p < 0.01)” (revised manuscript page 8, lines 191-193).

Comment 4: It is important to put in a statement to emphasize the allele frequency differences of this SNP in different populations in general.

Answer: The MAF of the “G” allele of rs700651 showed an approximate 2-fold difference in both European-ancestry and East-Asian populations (MAF = 0.2674 and 0.4614, respectively), confirmed in the 1000 Genome Phase 3 reference panel. Despite the fact that the allele frequency difference in the general populations and a potential ethnic difference in human genome structures were present, we demonstrated a disease-specific genetic similarity in the ethnic populations tested, which showed the same effect direction of this variant susceptibility to IA (Discussion section page 9, lines 237-242). The population-based MAFs were calculated by using the “LDproxy” package of the LDlink program (https://ldlink.nci.nih.gov/).

Reviewer 2 Report

Intracranial aneurysm (IA) is a serious medical problem as it is assessed that on average one in 25 individuals can be at risk of IA. Although in many cases IA is asymptotic, it may lead to subarachnoid hemorrhage (SAH), which is the reason of a significant immediate and delayed mortality among IA patients and serious disability. Therefore, IA prevention and early detection can limit mortality associated with SAH. That is why addressing the problem of association of IA susceptibility with genetic variability is justified. The authors performed an updated systematic review of genome-wide association studies of the G allele of the rs700651 polymorphism and IA occurrence in different ethnic populations. The calculation methods they applied are adequate to the subject they undertook. Authors’ final conclusion that the common variant of rs700651 may be involved in mechanism of pathogenesis of IA is moderately supported by the data they obtained showing genome-wide expression in some human tissues with a strong genome-wide significance threshold. However, the statement that this variant may be crucial for regulatory functions is rather vague. “Regulatory functions” of what? These study has several limitations and weaknesses, but the authors are aware of them and discuss them. This is rather well-written manuscript and I have only some minor remarks.

Author Response

Comment 1: Intracranial aneurysm (IA) is a serious medical problem as it is assessed that on average one in 25 individuals can be at risk of IA. Although in many cases IA is asymptotic, it may lead to subarachnoid hemorrhage (SAH), which is the reason of a significant immediate and delayed mortality among IA patients and serious disability. Therefore, IA prevention and early detection can limit mortality associated with SAH. That is why addressing the problem of association of IA susceptibility with genetic variability is justified. The authors performed an updated systematic review of genome-wide association studies of the G allele of the rs700651 polymorphism and IA occurrence in different ethnic populations. The calculation methods they applied are adequate to the subject they undertook. Authors’ final conclusion that the common variant of rs700651 may be involved in mechanism of pathogenesis of IA is moderately supported by the data they obtained showing genome-wide expression in some human tissues with a strong genome-wide significance threshold. However, the statement that this variant may be crucial for regulatory functions is rather vague. “Regulatory functions” of what? These study has several limitations and weaknesses, but the authors are aware of them and discuss them. This is rather well-written manuscript and I have only some minor remarks.

Answer: Per your recommendation, we described the potential regulatory functions of the 2q33.1 region tagging the rs700651, including the positive regulation of translational initiation (BOLL) and the regulation of synaptic transmission (PLCL1) by adding a discussion on the gene ontology of several candidate genes as shown below.

This meta-analysis highlights how an increase in the statistical power of rs700651 mapped to the 2q33.1 region played an important role in identifying the regulatory mechanisms, including the positive regulation of translational initiation (BOLL) and the regulation of synaptic transmission (PLCL1) related to IA susceptibility (revised manuscript page 9 and lines 255-258).

Round 2

Reviewer 1 Report

The authors addressed all the comments that I have suggested.